# Sintering Temperature Effect of Near-Zero Thermal Expansion Mn_3_Zn_0.8_Sn_0.2_N/Ti Composites

**DOI:** 10.3390/ma16175919

**Published:** 2023-08-29

**Authors:** Yongxiao Zhou, Lianyu Zhang, Jinrui Qian, Zhiying Qian, Baoxin Hao, Qiang Cong, Chang Zhou

**Affiliations:** 1State Key Laboratory for Advanced Metals and Materials, University of Science and Technology Beijing, Beijing 100083, China; 20b909024@stu.hit.edu.cn (Y.Z.); jinruiqian2022@163.com (J.Q.); 2School of Materials Science and Engineering, Harbin Institute of Technology, Harbin 150001, China; g318934@163.com; 3Institute of Spacecraft System Engineering, China Academy of Space Technology, Beijing 100083, China; everestqq2015@sina.com (Z.Q.); haobaox@buaa.edu.cn (B.H.); qiangcong@sohu.com (Q.C.)

**Keywords:** negative thermal expansion, zero thermal expansion, metal matrix composites, mechanical properties

## Abstract

Metal matrix composites with near-zero thermal expansion (NZTE) have gained significant popularity in high-precision industries due to their excellent thermal stability and mechanical properties. The incorporation of Mn_3_Zn_0.8_Sn_0.2_N, which possesses outstanding negative thermal expansion properties, effectively suppressed the thermal expansion of titanium. Highly dense Mn_3_Zn_0.8_Sn_0.2_N/Ti composites were obtained by adjusting the fabrication temperature. Both composites fabricated at 650 °C and 700 °C exhibited NZTE. Furthermore, finite element analysis was employed to investigate the effects of thermal stress within the composites on their thermal expansion performance.

## 1. Introduction

Negative thermal expansion (NTE) [1] particle-reinforced metal matrix composites (MMCs) are well known for their low thermal expansion [2]. Furthermore, these composites exhibit exceptional electrical and mechanical properties [3], which have captivated the attention of researchers. Previous studies have predominantly focused on MMCs reinforced with ZrW_2_O_8_ [4]. However, the capacity to inhibit the thermal expansion of metals is achievable only at a high volume fraction, mainly due to the limited NTE of ZrW_2_O_8_ (−8.1 ppm/K) [5]. In addition, ZrW_2_O_8_ possesses the characteristic of easy decomposition [6]. Thus, it is a challenge to achieve high-performance NZTE ZrW_2_O_8_-reinforced composites.

For aerospace applications, it is preferable for materials to demonstrate NZTE within the temperature range of −20 to 60 °C [7]. Anti-perovskite manganese nitride compounds (Mn_3_AN) exhibit significant isotropic NTE (about −20.0~−40.0 ppm/K) [8] in this temperature range among all NTE materials. Furthermore, it shows excellent electrical [9] and mechanical properties [10]. As a result, Mn_3_AN can effectively suppress the thermal expansion of the metal matrix while maintaining the electrical and mechanical performance of the composites even at low volume fractions [3,11]. Mn_3_AN is known for its ability to control NTE properties through chemical doping [12]. The NTE performance of Mn_3_ZnN is stranger than that of Mn_3_CuN and Mn_3_GaN. For example, Mn_3.1_Zn_0.5_Sn_0.4_N shows NTE of α = −45.2 ppm/K at 313~354 K (Δ*T* = 41 K) [10], whereas Mn_3_Cu_0.5_Sn_0.5_N shows NTE of α = −21.1 ppm/K at 282~322 K (Δ*T* = 40 K) [13], and Mn_3_Ga_0.5_Sn_0.5_N shows NTE of α = −8.1 ppm/K at 418~454 K(Δ*T* = 36 K) [14]. The superior NTE performance of Mn_3_ZnN makes it more suitable as a thermal expansion inhibitor [15].

Introducing negative thermal expansion reinforcement to metals provides an effective method for regulating thermal expansion. However, most of the research works focus on Cu [16,17,18] and Al [19,20]. Takenaka et al. fabricated different metal matrix composites (Al, Ti, Cu) with Mn_3_AN by spark plasma sintering with various volume fractions and all the composites achieved NZTE [21], which proves the effectiveness of Mn_3_AN. In addition, no more reports on Mn_3_AN-reinforced Ti matrix composites (TMC) can be found. There are still some issues that need to be solved in Mn_3_AN-reinforced TMCs: (1) achieving Mn_3_AN-reinforced TMCs and (2) characterizing the microstructure of TMCs and its impact on performance. These are of great significance for in-depth research of NZTE TMCs.

This study fabricated Mn_3_Zn_0.8_Sn_0.2_N/Ti composites using the spark plasma sintering (SPS) technique. Composites with varying densities, phase compositions, and microstructures were obtained by adjusting the fabrication temperature. The effects of these parameters on the thermal expansion and mechanical properties of the composites were investigated. Finite element analysis was conducted to calculate the internal thermal stresses of the composites under temperature variations. Furthermore, the influence of internal thermal stresses on the thermal expansion properties of the composites was elucidated. This work provides valuable insights into the development of fully dense and high-strength NZTE composites.

## 2. Materials and Methods

Pure Mn_3_Zn_0.8_Sn_0.2_N particles were synthesized with a mixture of Mn_2_N, Zn, and Sn powders in the stoichiometric ratio by vacuum solid-state reaction. The mixture was mixed evenly and then pressed into blocks under a pressure of 20 MPa for 5 min. These blocks were placed in a tube furnace and kept under vacuum at 800 °C for 48 h. The blocks were grinded with a mortar for 30 min after cooling to room temperature and then sifted out through a 300-mesh sieve.

The resulting Mn_3_Zn_0.8_Sn_0.2_N (abbreviated as MnZSN) powders (20–80 μm, average of 40 μm) were mixed with titanium powders (5–15 μm, average 10 of μm) at 30% volume fraction by a ball mill at 60 rpm for 12 h. The mixture was placed in a graphite mold and then sintered by SPS at a pressure of 30 MPa for 7 min at 650 °C, 700 °C, and 750 °C. Finally, after cooling to room temperature and demolding, the 30 vol.% Mn_3_Zn_0.8_Sn_0.2_N/Ti (abbreviated as MnZSN/Ti) composites were successfully obtained. Figure 1 illustrates the arrangement of the SPS experiment conducted at 700 °C. The mold was preheated to 400 °C before the initial stage. Meanwhile, the preform was subjected to a pre-pressure of 15 MPa for 3 min. Subsequently, pressure was gradually increased while maintaining constant temperature. After 6 min, the temperature was raised to 700 °C at a rate of 50 °C/min with the pressure increasing to 30 MPa. After maintaining temperature and pressure for 7 min, the sintering entered the cooling phase. Finally, the sintering process ends with a temperature drop to 400 °C while maintaining pressure.

The X-ray diffraction (XRD) results of composites were observed with an Empyrean diffractometer with Cu Kα radiation. The microstructure and elemental distribution were observed by a scanning electron microscope (SEM, SUPRA65, ZEISS, Jena, Germany) with an energy dispersive spectrometer (EDS, Oxford, UK). The thermal expansion curves were tested by a NETZSCH Dilatometer 402C at a heating rate of 5 °C/min in the temperature range from −75 to 50 °C.

## 3. Results and Discussion

### 3.1. Reinforcement Characterization

Reinforcement of high-purity particles serves as the foundation of composites with excellent performance. Therefore, we conducted a comprehensive analysis of the fabricated MnZSN. Figure 2a shows the surface morphology of MnZSN particles. It can be observed that the particles exhibit irregular polygons and have a relatively rough surface, which may lead to stress concentration within the composites. Figure 2b presents the particle size distribution of MnZSN. The particle sizes are predominantly distributed in the range of 20 to 80 μm with a median particle size of 40 μm. Figure 2c illustrates the phase analysis of MnZSN using XRD, which was compared with the standard PDF card. The comparison between the synthesized MnZSN particles and standard PDF card shows that there is strong agreement between the d-spacings, indicating that the processing produced the MnZSN particles desired.

Figure 3 shows the thermal expansion curve and instantaneous α of MnZSN. Based on distinct thermal expansion behaviors, the thermal expansion curve of MnZSN can be divided into three regions: the low-temperature near-zero thermal expansion region I (−75 °C to −45 °C), the negative thermal expansion region II (−45 °C to −5 °C), and the high-temperature positive thermal expansion region III (−5 °C to 50 °C). In region I, despite exhibiting positive thermal expansion, MnZSN demonstrates a relatively low coefficient of thermal expansion of 3.72 ppm/K. Transitioning into region II, the average α of MnZSN is −56.4 ppm/K. As indicated in Figure 3b, the instantaneous α can reach −120 ppm/K at −21 °C. This significant negative thermal expansion α ensures effective suppression of the thermal expansion of the Ti matrix. In region III, the average α is as high as 20.2 ppm/K.

Mechanical properties are also important indicators for evaluating the quality of materials. Figure 4 illustrates the bending and compression performance of MnZSN. It can be observed that MnZSN exhibits a deformation of only 0.24% and a bending strength of 233 MPa due to the brittleness of ceramics and internal defects in Figure 4a. Figure 4b demonstrates the favorable compression performance of MnZSN with a compression strength of 647 MPa and a compression strain of 11.8%.

### 3.2. Phase Composition and Microstructure

Figure 5 represents the XRD patterns of MnZSN/Ti composites under different sintered temperatures. Only the characteristic peak of MnZSN and Ti can be detected in the 650MnZSN/Ti and 700MnZSN/Ti. A large number of unknown diffraction peaks between 40~50° are detected in the 750MnZSN/Ti composite, indicating heavy chemical reaction. K. Takenaka prepared Mn_3.1_Zn_0.5_Sn_0.4_N/Ti composites at 650 °C and no chemical reaction was found, which was consistent with our work [21]. Our results further prove that fabrication temperature has a great influence on the phase composition of MnZSN/Ti composites. The chemical reaction becomes mild below 700 °C but accelerates when heating above 750 °C, which has not been reported before. We found that these diffraction peaks mainly belong to Ti–N compounds, including TiN and Ti_2_N. This indicates that Ti and MnZSN seize the N element in MnZSN during the reaction, thereby affecting the structure of MnZSN. This greatly weakens the negative thermal expansion performance of MnZSN, and even causes its loss.

Figure 6 depicts SEM images of the MnZSN/Ti samples at different fabrication temperatures. It can be seen that the composites consisted of three different structures in black, grey, and white. The grey areas correspond to the Ti matrix and the white ones are MnZSN reinforcements. The black regions within both the Ti and MnZSN are pores. Interestingly, the color of MnZSN switches to light grey for sintering at 750 °C due to the great chemical reaction. It can be found that the size of pores decreases as the sintering temperature rises from 650 °C to 750 °C. The density of 650MnZSN/Ti, 700MnZSN/Ti, and 750MnZSN/Ti were 4.72, 4.87, and 4.93 g/cm^3^. Despite the interface products, the theoretical density of the sample was determined to be 5.07 g/cm^3^. Accordingly, the relative density of 650MnZSN/Ti and 700MnZSN/Ti was 93.10% and 96.06%, respectively. However, it was hard to estimate the porosity of 750MnZSN/Ti composites due to a large number of chemical reaction products.

The relatively high density in the 700MnZSN/Ti sample was attributed to the plastic deformation of the Ti particles. Dirk attained dense titanium alloy by consolidating pure Ti powder via SPS at 700 °C/60 Mpa [22]. In this work, the rigid MnZSN may cause holes near the reinforcement and matrix. Although further increasing the sintering temperature leads to densification, the most suitable sintering temperature in the MnZSN/Ti system is 700 °C considering the chemical reaction. Adding low-melting-point alloys as sintering aids is a good way to densify the composite and is described in the next section.

Figure 7 and Figure 8 present the EDS mapping scan results of 700MnZSN/Ti and 750MnZSN/Ti, respectively. It can be observed from Figure 7 that the Ti element is separate from the Mn, Zn, Sn, and N elements in 700MnZSN/Ti. This indicates that there was no reaction between MnZSN and Ti within the composite. However, Figure 8 displays the different behaviors of elemental distribution in 750MnZSN/Ti. Although the Mn, Zn, and Sn elements are uniformly distributed within the MnZSN particles, the N element within MnZSN appears to be absent. The N element within the Ti matrix is significantly higher in concentration than that within MnZSN. In conjunction with the XRD results, it can be concluded that the N element of MnZSN in the 750MnZSN/Ti composite reacted with Ti to form Ti–N compounds, indicating significant reaction behavior within 750MnZSN/Ti.

### 3.3. Thermal Expansion

Figure 9a presents the temperature dependence of linear thermal expansion for the MnZSN/Ti composites. It can be observed that the overall strain of the composite sintered at 650 °C and 700 °C is 5.59 × 10^−4^ and 5.66 × 10^−4^ in the range of −75 to 50 °C, respectively. While the stain of Ti in this range was 11.2 × 10^−4^, the relatively low strain value means that the introduction of NTE MnZSN can effectively suppress the thermal expansion of Ti. Unfortunately, the overall strain of 750MnZSN/Ti increased to 10.51 × 10^−4^, almost twice as much as the composites sintered at 650 °C and 700 °C. This can be explained by the fact that severe interfacial reactions produce a large number of positive thermal expansion products as the sintering temperature is above 750 °C.

The CTE curve for 650MnZSN/Ti shows a stable line in section Ⅰ (−75 °C to −44 °C) with a low CTE of 2.84 ppm/K. Then, a shrink appears in section Ⅱ, and the CTE reduces to −3.50 ppm/K in the section from −44 °C to −15 °C, mainly due to the ferromagnetic phase transition of MnZSN reinforcement. After that, the curve rises again with a mean CTE value of 9.04 ppm/K in section Ⅲ, indicating the phase transition has been completed. As shown in Figure 9b, according to the rule of mixtures, the theoretical values of α for 30 vol.% MnZSN/Ti are α = 5.69 ppm/K at −75 °C~−45 °C, α = −14.68 ppm/K at −45 °C~−5 °C, and α = 11.46 ppm/K at −5 °C~50 °C. A comparison between experimental and theoretical values reveals that the thermal expansion performance of 650MnZSN/Ti in sections I and III is significantly lower than the theoretical values, while section II shows a noticeable deviation above the theoretical values. The difference in thermal expansion coefficients and temperature ranges may originate from the magnetic phase transition of Mn_3_Zn_0.8_Sn_0.2_N within the composites, which is discussed in the following sections.

The dilatometry result of 700MnZSN/Ti was similar to that of 650MnZSN/Ti. The mean CTE value was 3.24 ppm/K in the initial section I′ (−75 °C to −55 °C). In section Ⅱ′, additional abnormal NTE and a similar CTE value (−3.22 ppm/K) can be found in the range of −55 °C~−24 °C (section Ⅱ′), which was slightly higher than that of 650MnZSN/Ti. Then, the CTE curve has another stable section because the phase transition has finished. What should be noted is that although the CTE value remains almost the same, the NTE temperature window varies. As shown in Figure 9, it can be observed that the thermal expansion curve of pure MnZn_0.8_Sn_0.2_N also has three sections. As an antiferromagnetic phase, MnZn_0.8_Sn_0.2_N shows positive thermal expansion (3.72 ppm/K) in the temperature range from −75 to −45 °C. From −45 °C to −5 °C, MnZn_0.8_Sn_0.2_N undergoes an antiferromagnetic–paramagnetic (AFM–PM) phase transition at the Néel temperature (T_N_), leading to negative thermal expansion (α = −56.4 ppm/K) due to the magnet volume effect. After that, all phase transitions are completed and the MnZn_0.8_Sn_0.2_N shows positive thermal expansion again. The NTE temperature area in the reinforcement, 650MnZSN/Ti, and 700MnZSN/Ti are −45 °C to −5 °C (Δ*T* = 40 °C), −44 °C to −15 °C (Δ*T* = 30 °C), and −55 °C~−24 °C (Δ*T* = 31 °C). That is, the maximum T_N_ moves to a lower temperature. Chemical reactions and high thermal mismatch stress are the main factors affecting the thermal expansion performance of composites. However, based on our above results and previous reports, the chemical reaction between the MnZnSnN reinforcement and the Ti matrix is not active below 700 °C.

To further understand the nature of the variation of the magnetic transitions, the finite element method (FEM) was used to compute the magnitude of the residual thermal stress in the MnZSN/Ti composite. The volume fraction and diameter of the MnZSN were set as 30% and 48 μm, respectively. Materials parameters such as CTE (*α*), elastic modulus (*E*), and Poisson’s ratio (*v*) used in numerical computations were from experimental results and former references [23,24]. Here, *E_Ti_* = 115 GPa, *E_MnZSN_* = 158 GPa, *v_Ti_* = 0.33, and *v_MnZSN_* = 0.3. As shown in Figure 10, the mismatch stress in the interface was 125~150 MPa. The titanium matrix model uses ideal cubes and the Mn_3_Zn_0.8_Sn_0.2_N reinforcement in the model adopts an ideal spherical shape. The mesh is divided into three-dimensional solid elements. The matrix mesh is divided into eight-node hexahedral elements. The reinforcement mesh is divided using ten-node tetrahedral elements. We constrained the x, y, and z planes of the matrix and their intersections using placement.

It has been reported that the magnetic transition that occurs at T_N_ is the first-order magneto-structural transition, which is sensitive to thermal stress [25]. Daichi Matsunami reported that the pressure dependence of the Néel temperature (d_TN_/d_p_) of antiferromagnetic Mn_3_GaN was −65 K/GPa [26]. According to Figure 10, the thermal stress value caused by the mismatch of the thermal expansion coefficient can reach 125~150 MPa, which was the main reason for the movement of T_N_ and NTE regions in the composites. In our previous work, it was found that holes can release some residual stress [20]. Therefore, it can be concluded that compared with the 650MnZSN/Ti composite, the relatively low Tn of the 700MnZSN/Ti composite was mainly due to the denser structure prepared at high temperatures.

K. Takenaka achieved low thermal expansion in a 50 vol.% Mn_3.1_Zn_0.5_Sn_0.4_N/Ti composite (−2.2 ppm/K), while the CTE value was much higher than predicted due to chemical reaction [21]. In this article, the CTE expansion of 30 vol.% MnZSN/Ti composite is as low as −3.50 ppm/K. Lowering NTE fillers in metal matrix composites (MMCs) to achieve near zero thermal expansion or negative thermal expansion is crucial and can minimize the unfavorable reduction in the mechanical properties and thermal conductivity of MMCs. This is quite difficult for other NTE filler-reinforced MMCs such as ZrW_2_O_8_/Al, PbTiO_3_/Cu, and xLFCS/Cu composites. For example, roughly 70 vol.% ZrW_2_O_8_ was needed to realize near-zero thermal expansion in ZrW_2_O_8_/Al composites. In this work, the relatively low thermal expansion of Ti (8.4 ppm/K) and high NTE value (over −56 ppm/K) of high-purity MnZn_0.8_Sn_0.2_N make it possible.

### 3.4. Mechanical Properties

ZTE composites generally face the problem of low mechanical strength, which affects their application. Ti as the matrix can effectively improve the mechanical properties of ZTE composites due to its high strength. Figure 11a shows the bending strength of MnZSN/Ti composites. 700MnZSN/Ti exhibits the highest bending strength of 253 MPa, while the bending strengths of 650MnZSN/Ti and 750MnZSN/Ti are 150 MPa and 178 MPa, respectively. The significant strength reduction of these two composite materials compared to 700MnZSN/Ti can be explained as follows: for 650MnZSN/Ti, the density of the composite material is only 93.10%, which leads to the presence of many pores inside the composite, leading to early cracking and failure during the bending process; for the 750MnZSN/Ti composite, although increasing the fabrication temperature significantly increases the density of the composite, the subsequent interface reaction causes a large number of brittle phases to be generated in the composite. These brittle phases are concentrated at the interface between the matrix and reinforcement, which can also lead to the early formation of cracks and failure.

The Vickers hardness test is used to further characterize the mechanical properties of MnZSN/Ti composites, as shown in Figure 11b. The Vickers hardness of 650MnZSN/Ti, 700MnZSN/Ti, and 750MnZSN/Ti are 241 Hv, 260 Hv, and 317 Hv, respectively. The Vickers hardness of 700MnZSN/Ti is higher than that of 650MnZSN/Ti due to its higher relative density. The hardness of 750MnZSN/Ti significantly improved due to the reaction-generated TiN in the composite which has an extremely high Vickers hardness of 2200 Hv [27].

Figure 12 shows the bending fracture of the composites. Both 650MnZSN/Ti and 700MnZSN/Ti exhibit particle fracture phenomena. Compared to 700MnZSN/Ti, there are more visible pores in the matrix of 650MnZSN/Ti, which is the reason for its lower strength. At the fracture surface of 750MnZSN/Ti, the entire reinforcement particle is pulled out due to the brittle phase at the interface, which caused cracks that are prone to occur and propagate at the interface. This is the reason for its high density but also its decrease in strength.

## 4. Conclusions

In summary, we have synthesized Mn_3_Zn_0.8_Sn_0.2_N with negative thermal expansion. And 30 vol.% Mn_3_Zn_0.8_Sn_0.2_N/Ti composites were fabricated at different temperatures using spark plasma sintering. This work can be summarized as follows:Mn_3_Zn_0.8_Sn_0.2_N exhibits excellent negative thermal expansion performance, with a thermal expansion coefficient of −56.4 ppm/K in the range of −45 °C to −5 °C;Mn_3_Zn_0.8_Sn_0.2_N/Ti composites prepared at 700 °C demonstrate high density and low reactivity;The Mn_3_Zn_0.8_Sn_0.2_N/Ti composite fabricated at 700 °C exhibits excellent comprehensive performance with near-zero thermal expansion in the range of −55 °C to −24 °C (−3.22 ppm/K) and a bending strength of 253 MPa;Finite element analysis indicates significant thermal mismatch stress within the composites. This could influence the performance of Mn_3_Zn_0.8_Sn_0.2_N particles, leading to a displacement of the near-zero expansion temperature range within the composites compared to individual Mn_3_Zn_0.8_Sn_0.2_N particles.

## Figures and Tables

**Figure 1 materials-16-05919-f001:**
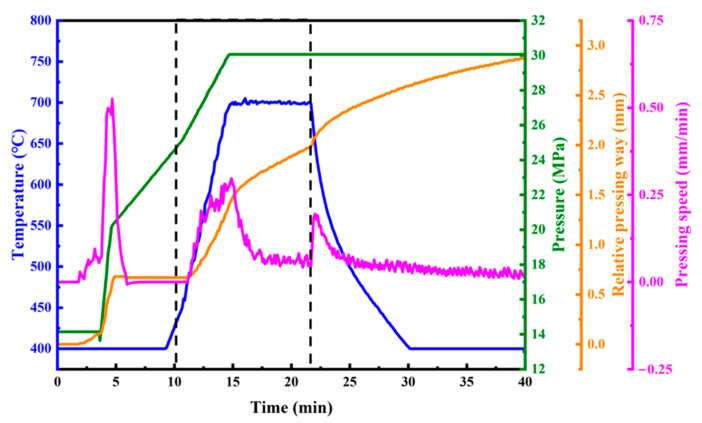
The schematic diagram of SPS sintering curves.

**Figure 2 materials-16-05919-f002:**
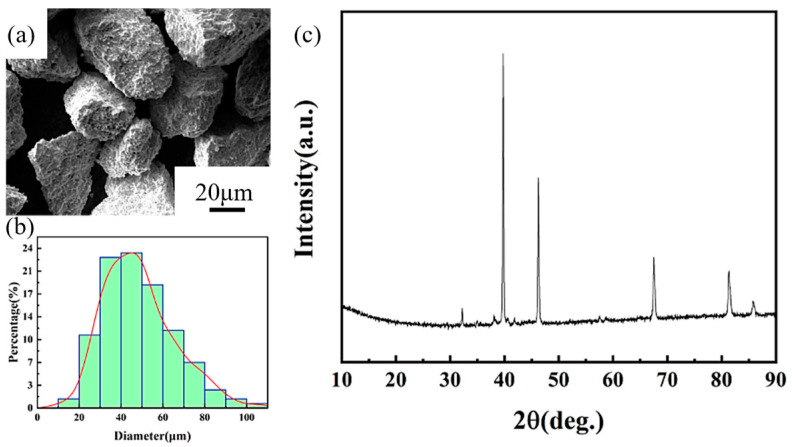
Characterization of MnZSN: (**a**) particle morphology; (**b**) particle size; (**c**) XRD results.

**Figure 3 materials-16-05919-f003:**
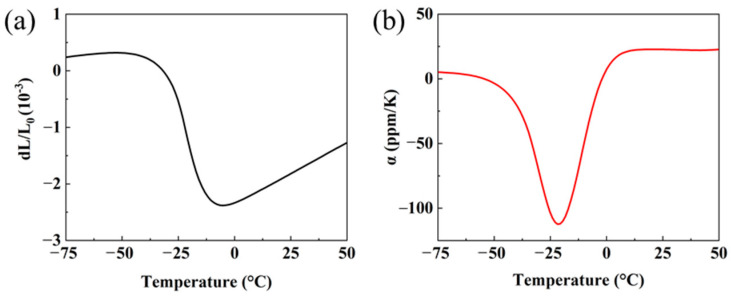
Thermal performance of MnZSN: (**a**) thermal curve; (**b**) instantaneous α.

**Figure 4 materials-16-05919-f004:**
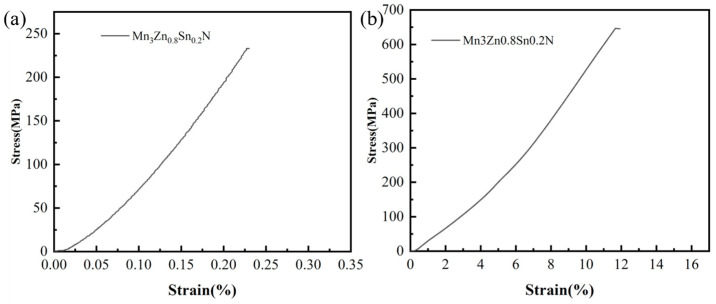
Mechanical properties of MnZSN: (**a**) bending stress; (**b**) compression stress.

**Figure 5 materials-16-05919-f005:**
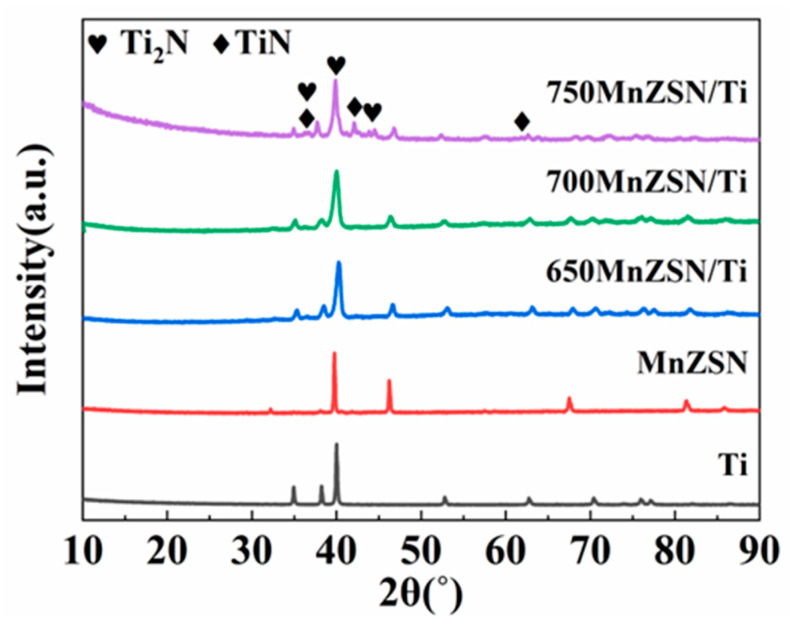
XRD results of MnZSN/Ti composites under different sintered temperatures.

**Figure 6 materials-16-05919-f006:**
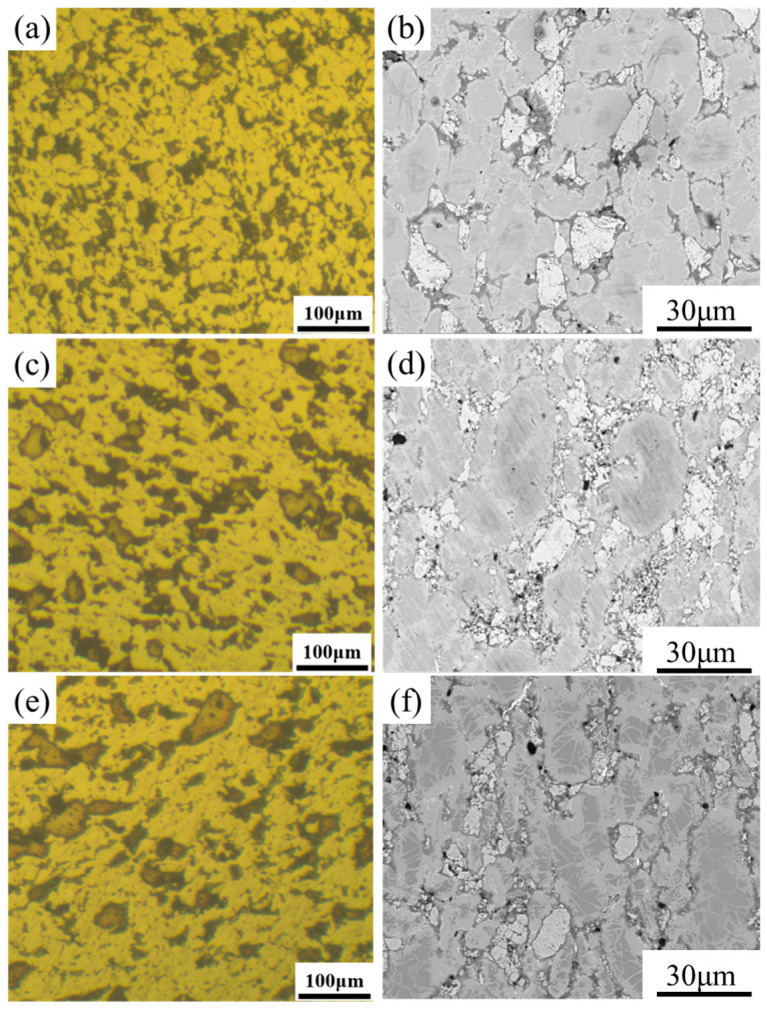
The SEM images of MnZSN/Ti composites with different fabrication temperatures: (**a**,**b**) 650 °C; (**c**,**d**) 700 °C; (**e**,**f**) 750 °C.

**Figure 7 materials-16-05919-f007:**
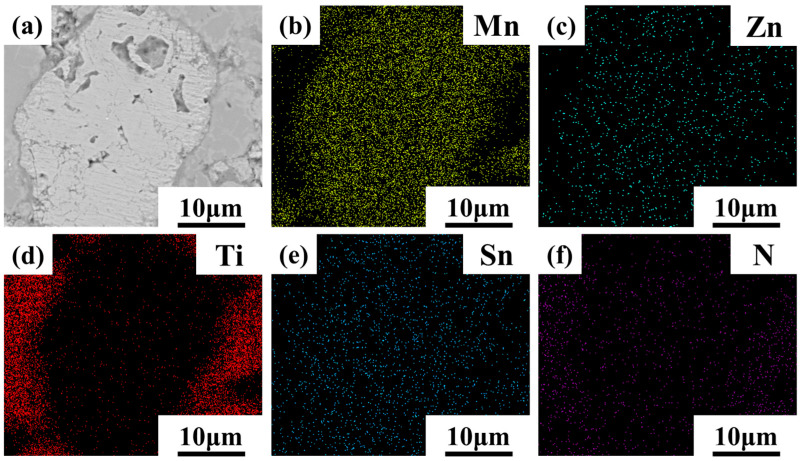
The EDS results of 700MnZSN/Ti. (**a**) microscopic structure; (**b**) Mn element distribution; (**c**) Zn element distribution; (**d**) Ti element distribution; (**e**) Sn element distribution; (**f**) N element distribution.

**Figure 8 materials-16-05919-f008:**
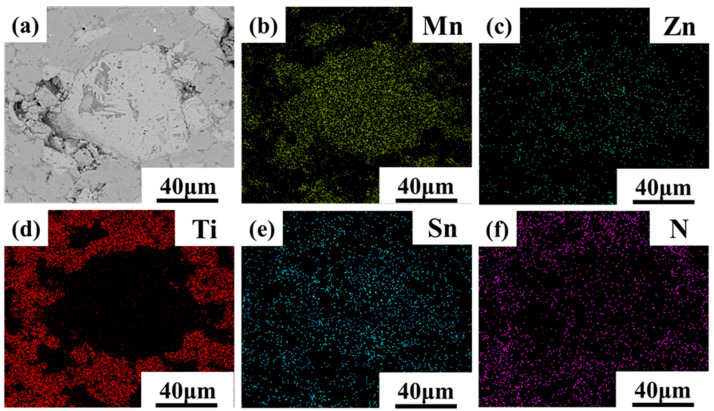
The EDS results of 750MnZSN/Ti. (**a**) microscopic structure; (**b**) Mn element distribution; (**c**) Zn element distribution; (**d**) Ti element distribution; (**e**) Sn element distribution; (**f**) N element distribution.

**Figure 9 materials-16-05919-f009:**
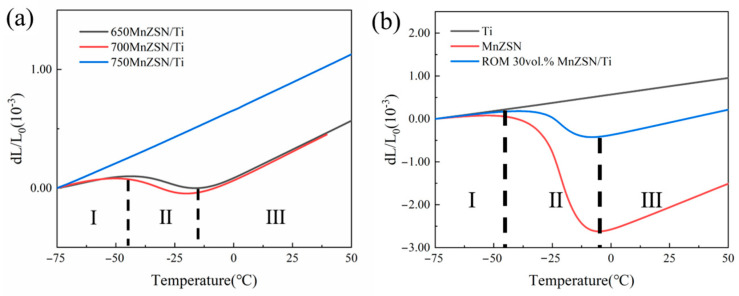
The thermal expansion curves of (**a**) experimental value of MnZSN/Ti; (**b**) MnZn_0.8_Sn_0.2_N, Ti, and theoretical MnZSN/Ti.

**Figure 10 materials-16-05919-f010:**
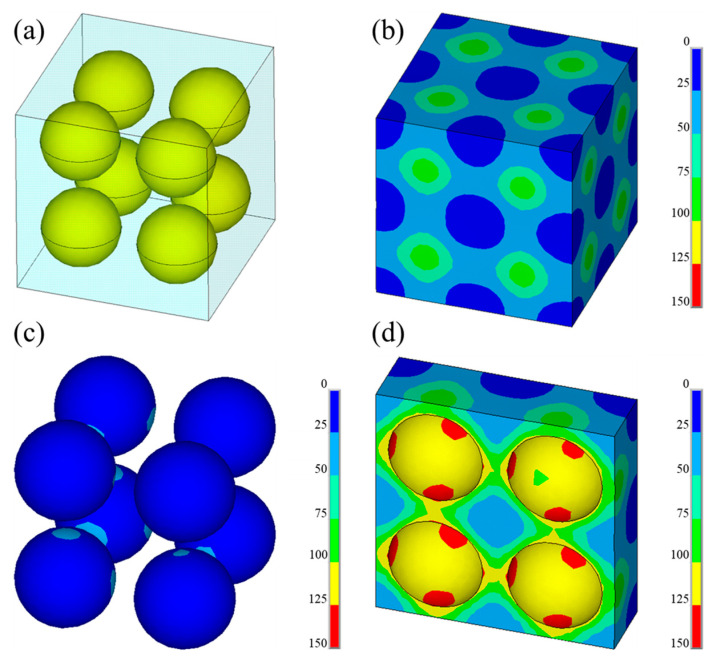
The von Mises stress distribution map of MnZSN/Ti composites: (**a**) cross section of composite; (**b**) surface of Ti matrix; (**c**) surface of MnZn_0.8_Sn_0.2_N particles; (**d**) cross section of Ti matrix.

**Figure 11 materials-16-05919-f011:**
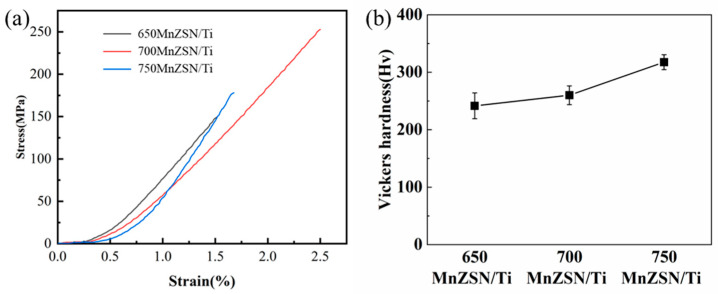
The mechanical properties of MnZSN/Ti composites: (**a**) bending strain–stress curves; (**b**) Vickers hardness.

**Figure 12 materials-16-05919-f012:**
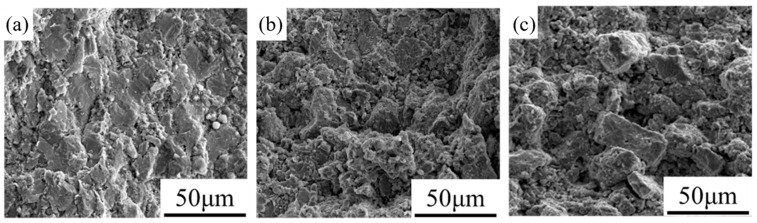
The bending strain–stress curves of MnZSN/Ti composites: (**a**) 650MnZSN/Ti; (**b**) 700MnZSN/Ti; (**c**) 750MnZSN/Ti.

## Data Availability

Data will be made available on request.

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
