# Peer review of "Sintering Temperature Effect of Near-Zero Thermal Expansion Mn3Zn0.8Sn0.2N/Ti Composites"

_materials, 2023, doi:10.3390/ma16175919_

Round 1

Reviewer 1 Report

The paper presents a study of near-zero thermal expansion of MnZSN/Ti composites.  The  paper frames the study as one of interest to aerospace (introduction), but separately high-precision (abstract) industries.  

The paper suffers from several issues, and can not be recommended for publication.  Firstly, the authors do frame this material as relevant for aerospace, yet point to the temperature range of -40 to +60°C - a range that has no citation, yet seems to 'magically fit' the data.  Neither aerospace engines, structures, or even space structures operate only in the reported temperature range.  Thus, the very premise is slightly suspect.  Secondly, aerospace depends upon 'performance' (which the authors note), but the measurements conducted do not seriously seek to understand properties.  They note that ZTE materials suffer from their strength, but I would contend that ductility is the more important variable.  Brittle materials can be statistically treated for strength, but even the culminating fraction composite that is 'optimum' in this study (30% MnZSN) is expected to have horribly low ductility, and would not be considered for many (any?) aerospace applications.  Further, the aging observations (700°C and above results in undesirable products) would further exclude this from being considered for the specified application, as HIP processes are quite often at 750°C or higher; and such treatments would be expected for this type of processing.

Thus, at a minimum, the paper introduces the material as relevant for aerospace, while then proceeding to present data that would make most aerospace engineers concerned about the material.  The technical details (other than those described below) may be interesting and worth presenting, but not at all with the framing presented.

The paper has other issues.  Regarding the materials and methods, the authors describe that the powder is consolidated into "blocks" and placed in a furnace, but come out of the furnace as "powder".  I suspect a step in the methods is missing - if it does come out as powder, it points to the extreme brittleness of the material.

The XRD DOES NOT show purity.  It shows d-spacings.  Those d-spacings may match a standard card, but no table of d-spacings from either the sample or the standard card is given.  Further, the interpretation is just incorrect - one can imagine that one impurity may lead to an increase in lattice parameter while another leads to a decrease in the lattice parameter.  This is why other methods for chemical analysis are pursued.

The SEM of powder does not show anything meaningful.  The authors had EDS at their disposal, but did not seem to use it in any meaningful way.  Even if they did, standard-based EDS would be a minimum, with ICP / GC or other chemical analysis would have been useful.

The observation of Region III MnZSN materials at 20.2 ppm/K is HIGHER than Ti at 1200K (14ppm/K).  So, the composite would be worse at high temperatures than the pure alloy.

References are missing throughout the article (e.g., Cao "attained dense titanium alloy" - line 136).  Which work?  Which Cao?  Was it "fully dense" or 90% dense or ???

The mismatch stress of 800 MPa is enormously high, and would lead to cracking/fracture immediately.

Reviewer 2 Report

The manuscript deals with the fabrication of Metal matrix composites with zero thermal expansion (ZTE). To fabricate such alloys the authors used Mn3Zn0.8Sn0.2N as filler material to suppress the thermal expansion of Titanium. They achieved reasonably dense composites. The fabrication process is clear and straightforward. The manuscript is legibly written. However, the manuscript cannot be published in Materials in its present form. The authors need to address certain comments before it can be considered for publication.

Comments:

1. The XRD data from fig. 4 shows that there might be an increase in the particle size of the composite sintered at 700 Deg. C. If there is no thermal expansion, then comment on the resultant particle size increase shown in fig. 4.

2.  Remove "?" from the fig.4

3. It is good to add elemental mapping in addition to fig. 5. This helps to clearly identify the various elements present in the composite.

4. Need to revisit English language usage in the manuscript.

5. Conclusion section need to be further improved.

There is a need to revisit English language usage throughout the manuscript.

Reviewer 3 Report

This manuscript “Sintering temperature effect of near-zero thermal expansion 2 Mn3Zn0.8Sn0.2N/Ti composites” is written well.

1.      Please give the snap shot of the experimental arrangement for sintering with explanation. Give the specification of the machine as well.

2.      Microhardness measurement could be further included.

3.      Details about the procedure to conduct the Finite Element Analysis should be included for the sake of readers.

4.      Conclusion is very brief. It could written point-wise highlighting the research results.

Round 2

Reviewer 1 Report

I appreciate the authors efforts at making corrections. There are still significant issues with this, but I can see merit in at least providing the data to the community.

The authors remain technically incorrect that XRD shows "purity" of a material.  Even if others claim in their paper or research that this is true, it simply is not true.  Purity is a chemical/compositional measurement.

As noted in my previous review, XRD shows d-spacings.  What the authors show here is an absence of other phases that may not be desired.  However, elements can both increase and decrease d-spacings.  So, a crystal that exhibits the desired d-spacing may or may not be "high purity".

The authors do not need to make this claim of purity in their paper.  They could state "The comparison between the two ... shows that there is strong agreement between the d-spacings of the two, indicating that the processing produced the particles desired" or something like that. 

Reviewer 2 Report

Please update Fig. 5 in the manuscript with the corresponding figure shown in response file. The manuscript can be accepted for publication after this update. I am satisfied with the author's responses for all the other questions raised earlier. 
